# Dynamics of *Borrelia burgdorferi*-Specific Antibodies: Seroconversion and Seroreversion between Two Population-Based, Cross-Sectional Surveys among Adults in Germany

**DOI:** 10.3390/microorganisms8121859

**Published:** 2020-11-25

**Authors:** Tom Woudenberg, Stefanie Böhm, Merle Böhmer, Katharina Katz, Niklas Willrich, Klaus Stark, Ronny Kuhnert, Volker Fingerle, Hendrik Wilking

**Affiliations:** 1Bavarian Health and Food Safety Authority, 85764 Oberschleissheim, Germany; Stefanie.Boehm@lgl.bayern.de (S.B.); merle.boehmer@lgl.bayern.de (M.B.); katharina.katz@lgl.bayern.de (K.K.); volker.fingerle@lgl.bayern.de (V.F.); 2ECDC Fellowship Programme, Field Epidemiology Path (EPIET), European Centre for Disease Prevention and Control (ECDC), 16973 Solna, Sweden; 3Postgraduate Training for Applied Epidemiology (PAE), Robert Koch Institute, 13353 Berlin, Germany; 4Institute of Social Medicine and Health Systems Research, Otto-von-Guericke-University, 39106 Magdeburg, Germany; 5Department for Infectious Disease Epidemiology, Robert Koch-Institute, 13353 Berlin, Germany; willrichn@rki.de (N.W.); starkk@rki.de (K.S.); wilkingh@rki.de (H.W.); 6Department for Epidemiology and Health Monitoring, Robert Koch-Institute, 13353 Berlin, Germany; kuhnertr@rki.de; 7German National Reference Centre for Borrelia, 85764 Oberschleißheim, Germany

**Keywords:** Lyme borreliosis, *Borrelia burgdorferi*, seroprevalence, seroconversion, seroreversion, tick-borne disease

## Abstract

Lyme borreliosis (LB) caused by *Borrelia burgdorferi* spp. is the most common human tick-borne disease in Europe. Although seroprevalence studies are conducted in several countries, rates of seroconversion and seroreversion are lacking, and they are essential to determine the risk of infection. Seropositivity was determined using a two-step approach—first, a serological screening assay, and in the event of a positive or equivocal result, a confirmatory immunoblot assay. Seroconversion and seroreversion rates were assessed from blood samples taken from participants included in two nation-wide population-based surveys. Moreover, the impact of antigen reactivity on seroreversion rates was assessed. The seroprevalence of antibodies reacting against *B. burgdorferi* spp. in the German population was 8.5% (95% CI 7.5–9.6) in 1997–99 and 9.3% (95% CI 8.3–10.4) in 2008–2011. Seroprevalence increased with age, up to 20% among 70–79 year-olds. The age-standardized seroprevalence remained the same. The yearly seroconversion rate was 0.45% (95% CI: 0.37–0.54), and the yearly seroreversion rate was 1.47% (95% CI: 1.24–2.17). Lower levels of antibodies were associated with seroreversion. Participants with a strong response against antigen p83 had the lowest odds on seroreversion. Given the yearly seroreversion rate of 1.47% and a seroprevalence up to 20% in the oldest age groups, at least 20% of the German population becomes infected with *B. burgdorferi* in their lifetime. The slight increase in seroprevalence between the two serosurveys was caused by an aging population.

## 1. Introduction

Lyme borreliosis (LB) is a disease caused by spirochaetal bacteria belonging to the *Borrelia burgdorferi* sensu latu complex (thereafter referred to as *B. burgdorferi*) transmitted by Ixodes spp ticks [1]. In North America, *B. burgdorferi* sensu stricto and *B. mayonii* are the only species causing LB, whereas in Europe, at least five species cause LB, namely *B. burgdorferi* sensu stricto, *B. afzelii*, *B. bavariensis*, *B. garinii*, and *B. spielmanii*. Clinical presentations can be observed in several organs [2]. Erythema migrans (EM), an early local infection of the skin, is the most common presentation and occurs several days to weeks after a tick bite [2,3]. The dissemination of *B. burgdorferi* can lead to more severe manifestations such as neuroborreliosis, Lyme arthritis, and carditis. The surveillance of Lyme borreliosis is problematic due to the fact that a clear stand-alone meaningful laboratory diagnostic test is not available, the need for notification among physicians, and the predictable effects of underreporting and diagnostic inaccuracy.

Yearly incidence of reported cases of LB is around 30 cases per 100,000 inhabitants per year in the US [4] and ranges from 0.6 in Ireland to 300 in Austria per 100,000 inhabitants in Europe [5]. Even within countries, incidence of LB is heterogeneous. In Germany, the annual reported incidence ranges considerably between districts from 0.5 cases to 200 cases per 100,000 inhabitants [6]. Data are subject to varying degrees of underreporting and underascertainment especially of mild cases. In the US, the total LB cases was estimated to be ten times higher than that found through regular surveillance [7]. In addition, in some countries, LB is not a mandatory notifiable disease, or notification is only partly mandatory [6,8,9]. From 2007 until 2017, the number of reported cases was stable (approximately 5000 cases per year) in five states in Germany [6].

A higher LB incidence is often observed in regions with a high density of the vector *Ixodes* spp. A typical habitat for the transmission of *B. burgdorferi* spp. usually provides sufficient humidity for the development and survival of ticks, and it supports a range of potential vertebrate reservoir hosts, such as small mammals [1]. A higher incidence in forestry areas is also reflected in higher seroprevalences of antibodies against *B. burgdorferi*. Forestry workers have high seropositivity rates ranging from 8% to 22% in Europe [10,11,12,13,14]. Population-based estimates of the seroprevalence among the general population in European countries varied around 4% in Norway, Belgium, and Finland (REF). In Germany, seroprevalence was 4% in children [15] and 9% in adults [16].

Seropositivity in these seroprevalence studies was assessed by a two-step approach, which is also recommended for the microbiological confirmation of LB infections in the clinical setting [17]. Two age groups with an increased risk were identified, namely children below 18 years and in particular adults above 60. Explanations for these increased risks need more investigation but could be due to behavioral factors (e.g., leisure time) and due to temporal risk differences for birth cohorts. For a more precise assessment of the infection pressure (incidence), a representative study with two measurement times for the participants is required. Using the sera from the preceding Federal Health Survey (BGS 98) as well as the German Health Interview and Examination survey for Adults (DEGS 2008–2011), we were able to do so. By measuring the antibodies between these two cross-sectional studies, it is possible to determine the incidence of new infections as well as the seroreversion rate.

The first objective of this study was to determine the incidence (seroconversion) of primary infections with *B. burgdorferi* in Germany. The second objective was to estimate rates of seroconversion and seroreversion, and their risk factors including demographic variables and antigens used in the immunoblot assay. Exploration of the association between antigen-specific antibodies and seroreversion potentially provides insights on the maturation of the antibody response and hence on the clinical manifestations of the infection [18,19].

## 2. Materials and Methods

### 2.1. Study Procedures

From 1997 until 1999, the first nationally representative health survey of adults between 18 and 79 years of age was conducted, which was called the German National Health Interview and Examination Survey 1998 (BGS98) [20]. From 2008 to 2011, the second nationally representative survey was conducted (German Health Interview and Examination survey for Adults (DEGS) [21]). The Robert Koch Institute collected blood samples in both surveys. These blood samples enabled the assessment of seropositivity for *B. burgdorferi*. Interviews of participants provided data on the health status. Estimates of the DEGS study have already been described [16]. Participants of BGS98 were also invited to participate in DEGS providing a subset of participants who were tested twice.

Both studies were conducted using two-stage stratified cluster sampling. Primary sampling units were municipalities, which were stratified by the region and degree of urbanization. Sampling probability of municipalities was dependent on the population size. Within municipalities, random samples of individuals, stratified by 10-year age groups, were taken. In both studies, eligible participants were aged between 18 and 79 years and permanent residents of Germany. All participants were required to provide a written informed consent. The ethical review board of the Charité University Hospital approved both studies. Further information on the study protocol, concept, and design of both studies are described in detail elsewhere [20,21].

### 2.2. Data Collection 

Participants went through a set of interviews and examinations, which included a standardized self-filled questionnaire and the collection of a venous blood sample. For this study, we used data on sex, age, place of residence, and socio-economic status. Our outcome of interest was seropositivity for *B. burgdorferi* infection. 

### 2.3. Serology

To ensure comparability, seropositivity was defined with the same screening assays in the same laboratory and complied with guidelines on the microbiological diagnosis of LB [22,23]. A two-step approach was used—first, a serological screening assay, and in the event of a positive or equivocal result, a confirmatory assay. Positive samples were defined as either (i) a positive ELISA result and an equivocal or positive result in the confirmatory assay, or (ii) an equivocal ELISA result and a positive result in the confirmatory assay.

The serological screening assay used an enzyme-linked immunosorbent assay (ELISA), which tested for *B. burgdorferi* IgG (Enzygnost Lyme link VlsE/IgG, Siemens Healthcare Diagnostics GmbH, Eschborn, Germany). This quantitative ELISA is based on a detergent extract from cultured *B. afzelii* (strain PKo) mixed with recombinant VlsE from *B. burgdorferi* s.s. (strain B31), *B. afzelii* (strain PKo), and *B. bavariensis* (strain PBi). We used a floating cut-off value automatically calculated by the alpha method to determine whether samples were positive, equivocal, or negative. Outcomes of ELISA are optical densities expressed in arbitrary units.

The confirmatory assay was a line immunoblot (Borrelia Europe plus TpN17 LINE, IgG, Virotech, Rüsselsheim, Germany). This test includes the purified antigens OspC (outer surface protein C), p83 (both from *B. afzelii* strain PKo), the recombinant antigens VlsE (Variable major protein-like sequence Expressed, from *B. burgdorferi* s.s. strain B31 and *B. garinii* strain IP90), p39 (BmpA), and DbpA (Decorin binding Protein A, from *B. garinii* strain PBr, *B. bavariensis* strain PBi, and *B. spielmanii* strain A14s). Detection of antibodies in the immunoblot is dependent on the strains and recombinant antigens used as antigen. Likewise, the immune response in infected patients can be directed to strain-specific *B. burgdorferi* spp proteins [24] and is also dependent on time since infection [17].

### 2.4. Statistical Analysis

To estimate representative seroprevalence estimates, we used the sampling weights provided by each of the studies and took into account the survey design. As age is a strong predictor of being seropositive [16], and the general population of Germany is becoming older, we also estimated the age-standardized seroprevalence of the second survey (DEGS) if the age distribution had remained the same since the first survey (BGS98) [25].

We estimated the seroconversion rate using the linked samples on the follow-up subset of participants after stratifying by years-between-sampling using the following formula on each stratum: pconvyear = 1 − (1 − pconvyearx)1/X, where pconvyear is the estimated probability of seroconverting per year and pconvyearx is the observed rate of seroconversion in the x years between the two time points of sampling. A constant rate of infection was assumed, and cycles of seroconversion and seroreversion in between the sampling were neglected. The seroreversion rate was estimated by dividing the total number of seroreversions by the average observation time of participants who tested positive in the first survey (BGS98) and were resampled in the second survey (DEGS). Confidence intervals for the yearly seroconversion and seroreversion rates were calculated using a bootstrap procedure based on resampling of the linked serum samples. The influence of different associated factors on seroconversion and seroreversion was determined using logistic regression models. We included the optical density in the first survey in the multivariable model for seroreversion to disentangle demographic factors from degree of antibody response. We first assessed the association with univariable logistic regression models and subsequently developed a multivariable model, which included sex, age, place of residence, population, and socio-economic status. For all analyses, we used R (version 3.6.0). We used R package “survey” to take into account the study design and sampling weights in our analyses [26].

## 3. Results

### 3.1. Participants

A total of 5510 participants were included from the BGS98 (1997–1999) survey and 6957 were included from the DEGS (2008–2011) survey. This represents 93% and 97% of the original study population, respectively. Of 2438 individuals, samples were tested for *B. burgdorferi*-specific antibodies in both surveys.

### 3.2. Trend Analyses between Two Cross-Sectional Studies

Among those measured in BGS98, 487 (8.8%, unweighted) tested positive (Appendix A) compared with 738 (10.6%, unweighted) in DEGS (Appendix A). Within both surveys, older age, being male, and living in south Germany were positively associated with seropositivity (Appendix A). Seroprevalence (weighted) increased to 20% (95% CI: 16.7–23.3) among 70–79 year-olds (Figure 1A). The overall weighted seroprevalence increased from 8.5% (95% CI 7.5–9.6) in 1997–1999 to 9.3% (95% CI 8.3–10.4) in 2008–2011. Whereas the seroprevalence increased up to 6% in most birth cohorts (Figure 1B), we observed a diverse change in the seroprevalence within age groups between the two surveys (Appendix A, Figure 1A). The age-standardized seroprevalence was 9.2% in BGS98 and 9.3% in DEGS. 

### 3.3. Rates of Seroconversion and Seroreversion

A total of 2438 participants were enrolled in both studies. The majority remained either positive (n = 177, 7%) or negative (n = 2,106, 86%) in both studies, and 6% changed (n = 155). Among those who tested negative in 1997–1999 (n = 2,223), 117 tested positive in 2008–2011 (5%), and among those that tested positive in 1997–1999 (n = 215), 38 were negative in 2008–2011 (18%) (Table 1).

The total observation time among all participants between the two studies was 29,192 person-years and on average 12 years. The yearly seroconversion rate was 0.45% (95% CI: 0.37–0.54) and the yearly seroreversion was 1.47% (95% CI: 1.24–2.17).

### 3.4. Seroconversion 

A higher proportion of males seroconverted (8.1%; (95% CI: 6.7–9.9)) compared with females (2.6%; (95% CI: 1.8–3.7)). The median age was 47 for participants who seroconverted and 44 for participants who tested negative in both studies. In univariable regression models, seroconversion was highest in participants resident in southeast Germany, particularly in smaller towns with a population below 50,000. Participants resident in federal states Baden-Württemberg, Bavaria, Saxony, and Thuringia had higher odds on seroconversion than participants from other federal states in Germany. The odds of seroconversion decreased with increasing urbanization. In the multivariable regression models, gender was the only significant variable, and males had 3.55 times the odds (95% CI: 2.33–5.57) of seroconversion compared with females.

### 3.5. Seroreversion

Seroreversion was characterized by a low optical density in in the first survey (Figure 2), being female and of younger age (Table 2). An increase of 1 year in age was associated with 0.97 times the odds of seroreversion in the multivariable logistic regression. Therefore, the likelihood of seroreversion decreased with older age. Males had lower odds of seroreversion compared with females (0.21, 95% CI: 0.08–0.51). For every 2-fold increase in the optical density, the odds of seroconversion were 0.39 times lower (0.24–0.55).

The geometric mean of the optical density of all sera of the first survey (BGS98) was 1.05 in comparison to 0.76 of all sera of the second survey (DEGS). The geometric mean of participants that tested positive in both serosurveys was 1.22 in the first and 0.99 in the second survey. The geometric means of the optical density of participants who tested positive in the first survey but negative in the second survey was 0.51 in the first and 0.21 in the second survey. Participants who tested positive in the first survey had an average decline in the optical density of 0.41 (in this analysis, we excluded participants who showed a two-fold increase in the optical density, as we presumed re-exposure to *B. burgdorferi* in this group). This decline was not different between participants who tested positive in both serosurveys and participants that experienced seroreversion (*p*-value = 0.578).

In addition, we assessed whether the presence of antibodies directed to a particular protein of *B. burgdorferi* was associated with seroreversion. In the univariable models, antibodies directed to antigens OspC, p39, p58, p83, and DbpA were associated with testing positive in 2008–2011 (Table 3). For example, seropositive individuals in 1997–1999 who had antibodies directed against p39 had 0.49 reduced odds of testing negative in 2008–2011 compared with individuals without antibodies reacting against this antigen. An increased number of antibodies directed against antigens was associated with decreased odds of seroreversion. When we included all immunoblots in one model, including the optical density, only individuals with antibodies against antigen p83 had statistically significant reduced odds of seroreversion (0.27, 95% CI 0.06–0.84); i.e., having antibodies directed specifically to protein p83 in the first survey was associated with seropositivity at the second measurement in 2008–2011, regardless of optical density and the presence of antibodies directed to other antigens listed in the multivariable model.

## 4. Discussion

The study provides important data for assessing the distribution of disease burden for Lyme borreliosis in Germany. For the first time, it enables a quantification of the incidence of *B. burgdorferi* infections. The aging German population explained the increase of 0.8% in the weighted seroprevalence as age-standardized estimates were similar. A stable seroprevalence is in line with notification data, the annual reported cases were around 5000 between 2007 and 2017 in five federal states of Germany [6].

Risk factors for seropositivity resembled those for seroconversion and were negatively associated with seroreversion. Male sex, older age, living in rural areas, and living in Southern Germany were all associated with seropositivity, seroconversion, and seroreversion [16]. This suggests that individuals who have an increased risk of being seropositive are also at an increased risk of becoming re-infected or having longer lasting antibody levels [27]. Persistent infections or relapses of LB seem to play a minor role, as two studies from the US showed that almost all patients who were treated adequately for a previous *B. burgdorferi* infection experienced a re-infection rather than a persistent infection or relapse of LB [28,29].

Seroprevalence estimates increased slightly across almost all birth cohorts by 2–6%, whereas age-specific seroprevalence estimates did not seem to increase systematically. This would suggest that the risk of infection is approximately the same in each birth cohort and that the yearly conversion rate of 0.45% applies to all birth cohorts. However, age-specific seroprevalence estimates seemed to be stable around 5–6% for adults younger than 50 and increased to 20% among 70–79-year-olds. Surveillance data from Germany show a bimodal distribution in the incidence of reported LB with a higher incidence in children (5 to 9 years) and adults between 50 and 80 years of age [6]. The differing reported incidence and measured seroprevalence could be caused due to different manifestations of disease and risk behaviors by age group.

The strongest predictor of seroreversion was the level of antibodies measured at first measurement. Participants with low levels of antibodies had higher odds of seroreversion. Immunity seems to wane over time, as observed elsewhere [30,31]. The waning immunity was reflected in the yearly seroreversion rate of 1.47%. Participants with antibodies directed against antigen p83 had the highest probability of remaining seropositive. Hauser et al. showed that the presence of a range of bands, including p39, p58, and p83, were all more frequent in sera from patients with neuroborreliosis and late LB compared with sera of patients with Erythema migrans [18,19]. Thus, detecting p83-specific antibodies in sera is a strong indication of a maturated immune response as a result of e.g., disseminated or multiple infections [30,31].

A strength of our study is the use of the two-step testing approach. Using this approach, which is also widely used in the clinical setting, makes our results of clinical value. However, the two-step approach has been criticized for its low sensitivity in early infection, technical complexity, and subjective interpretation when scored by visual examination [32,33]. However, the two-step approach has been the standard for laboratory diagnosis of LB in the last two decades due to the combination of the sensitive EIA and the Western immunoblot to increase specificity, but new serodiagnostic testing for LB is near-at-hand [32].

A limitation of our study is that seropositivity is not a direct reflection of disease. *B. burgdorferi* infections are often asymptomatic but may nevertheless result in antibody production [34]. On the other hand, the most frequent manifestation of Lyme borreliosis, Erythema migrans, is often seronegative, especially when treated early with antibiotics [35], while more severe infections are associated with a high proportions of seropositivity [30,36]. For a clearer interpretation of serological data, enabling asymptomatic infections to be distinguished from clinical disease, further research should be focused on the persistence of antibodies and in particular the study of antigen-specific antibodies in relation to clinical outcomes.

Up to 20% of the German population developed *B. burgdorferi*-specific antibodies in their lifetime. Considering the yearly seroreversion rate of 1.47%, the percentage of infections among the Germany population can be considered an underestimate. The proportion that developed clinical illness and sequelae among these 20% is unclear but could be considerable as more severe infections are associated with long-lasting antibodies. In the absence of a licensed vaccine, enhanced public health intervention, especially targeted to risk groups identified in our study, could reduce this burden, such as communicating that LB can be prevented by avoiding tick-infested environments, using tick repellents on clothing and skin, inspection of the entire skin surface after exposure for early tick removal, and early treatment after infection.

The results also improve the ability of stakeholders in public health to provide information on this disease in connection with the frequent inquiries from the public. It provides meaningful results on the extent to which a vector-borne disease changes over time in Germany and thus also makes a valuable contribution to the scientific discussion regarding the effects of climate change on public health in Germany.

## Figures and Tables

**Figure 1 microorganisms-08-01859-f001:**
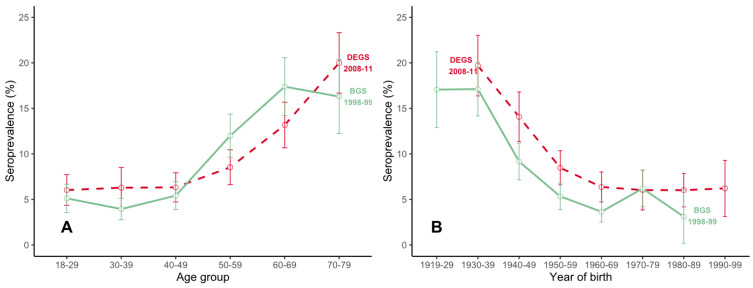
Weighted prevalence of burgdorferi antibodies by age group (**A**) and year of birth (**B**). In Germany, 1997–1999 (in green), and 2008–2011 (in red).

**Figure 2 microorganisms-08-01859-f002:**
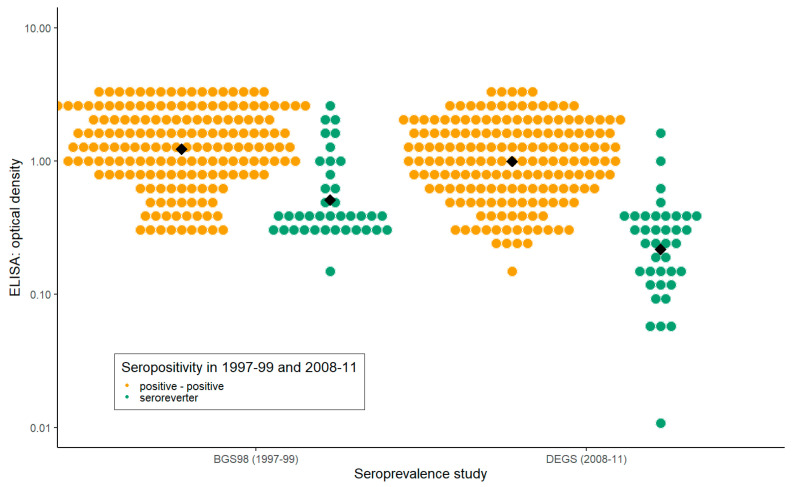
Optical density values of individuals who tested positive in both serosurveys (in yellow) in comparison to individuals who tested positive in 1997–1999 and negative in 2008–2011 (seroreversion, in green). The black squares visualize the geometric mean.

**Table 1 microorganisms-08-01859-t001:** Predictors of seroconversion of *Borrelia burgdorferi*-specific antibodies among participants of two population-based serosurveys in Germany, 1997–1999, and 2008–2011 (n = 2223).

Characteristic	Number of Seronegative Participants in BGS98	% Seronegative Participants in BGS98	Number of Seroconversions	% of Seroconversions	*p*-Value *	Univariable Analyses		Multivariable Analyses	
	**N = 2223**		**N = 117**			**OR**	**95% CI**	**OR**	**95% CI**
*Sex*					<0.01				
Female	1155	52	30	26		Ref			
Male	1068	48	87	74		3.33	2.2–5.16	3.55	2.33–5.57
**Age °**					0.42	1.01	0.99–1.02	1.01	0.99–1.02
18–29	277	13	11	9					
30–39	587	26	30	26					
40–49	542	24	31	27					
50–59	529	24	29	25					
60–69	241	11	13	11					
70–79	47	2	3	3					
**Place of residence ^§^**					0.05				
Baden-Württemberg	225	10	14	12		Ref			
Bavaria	244	11	16	14		1.06	0.5–2.25	1.01	0.47–2.21
Central	260	12	10	9		0.60	0.26–1.38	0.58	0.24–1.33
Northwest	318	14	15	13		0.75	0.35–1.59	0.80	0.37–1.75
North Rhine-Westphalia	357	16	10	9		0.43	0.18–0.99	0.47	0.20–1.10
East (north)	416	19	20	17		0.76	0.38–1.57	0.73	0.35–1.54
East (south)	403	18	32	27		1.30	0.69–2.57	1.28	0.66–2.59
**Population**					0.03				
<5000	550	25	39	33		Ref			
5,000–50,000	840	38	48	41		0.79	0.51–1.23	0.96	0.59–1.54
50,000–500,000	562	25	19 (16)	16		0.46	0.26–0.79	0.57	0.31–1.03
>500,000	271	12	11	9		0.55	0.27–1.06	0.79	0.36–1.61
**Socio-economic status**					0.50				
High	573	26	28	24		Ref			
Middle	1374	63	78	67		1.17	0.76–1.85	1.30	0.83–2.09
Low	249	11	10	9		0.81	0.37–1.65	0.85	0.38–1.76

* For comparison of seroconversion rates, we used the Chi2-test for distributions and a two-sample t-test for age. ^§^ Central comprises federal states Hessen, Rhineland-Palatinate, and Saarland; Northwest: Bremen, Hamburg, Lower Saxony, and Schleswig-Holstein; East (north): Berlin, Brandenburg, Mecklenburg-Vorpommern, and Saxony-Anhalt; East (south): Saxony, and Thuringia. ° Age was included as a continuous linear variable in the logistic regression models. Ref: reference category.

**Table 2 microorganisms-08-01859-t002:** Predictors of seroreversion of *Borrelia burgdorferi*-specific antibodies among participants of two population-based serosurveys in Germany, in 1997–1999 and 2008–2011.

Characteristic	Number of Seropositive Participants in 1997–1999	% of Seropositive Participants in 1997–1999	Number of Seroreversion	% of All Seroreversions	*p*-Value *	Univariable Analyses		Multivariable Analyses	
	**N = 215**		**N = 38**			**OR**	**95% CI**	**OR**	**95% CI**
**Sex**					<0.01				
Female	72	33	21	55		Ref	-	-	-
Male	143	67	17	45		0.33	0.16–0.67	0.21	0.08–0.53
**Age**					0.31	0.99	0.96–1.01	0.97	0.94–1.00
18–29	18	8	6	16					
30–39	26	12	6	16					
40–49	30	14	3	8					
50–59	75	35	11	29					
60–69	56	26	11	29					
70–79	10	5	1	3					
**Optical density (1997–1999) ^$^**					<0.01	0.38	0.25–0.54	0.39	0.24–0.55
<0.582	54	25	27	71					
0.582–1.2	54	25	5	13					
1.2–2.08	53	25	4	11					
>=2.08	54	25	2	5					
**Place of residence ^§^**					0.16				
Baden-Württemberg	32	15	7	18		Ref	-	-	-
Bavaria	32	15	5	13		0.66	0.18–2.34	0.49	0.09–2.35
Central	35	16	6	16		0.74	0.21–2.51	0.42	0.09–1.77
Northwest	28	13	2	5		0.27	0.04–1.27	0.25	0.03–1.49
North Rhine-Westphalia	19	9	6	16		1.65	0.45–6.01	1.62	0.29–9.24
East (north)	28	13	8	21		1.43	0.44–4.73	1.74	0.42–7.39
East (south)	41	19	4	11		0.39	0.09–1.42	0.29	0.06–1.33
**Population**					0.91				
<5000	62	29	10	26		Ref	-	-	-
5000–50,000	88	41	17	45		1.25	0.53–3.03	0.63	0.20–1.96
50,000–500,000	51	24	8	21		0.97	0.34–2.67	0.33	0.07–1.35
>500,000	14	7	3	8		1.42	0.28-5.59	0.45	0.06–3.02
**Socio-economic status**					0.57				
High	56	26	11	29					
Middle	132	62	21	55		0.77	0.35–1.79	0.49	0.17–1.39
Low	25	25	6	16		1.29	0.4–3.93	0.90	0.21–3.60

* For comparison of seroreversion rates, we used the Chi2-test for distributions and a two-sample t-test for age and log transformed optical density score. ^$^ Optical density is included linearly in the model. In order to be interpretable, we log-transformed optical density, so that a doubling in the optical density is 0.38 times the odds of seroreversion. Age is also included as a linear variable in the model. ^§^ Central comprises federal states Hessen, Rhineland-Palatinate, and Saarland; Northwest: Bremen, Hamburg, Lower Saxony, and Schleswig-Holstein; East (north): Berlin, Brandenburg, Mecklenburg-Vorpommern, and Saxony-Anhalt; East (south): Saxony, and Thuringia. Ref: reference category.

**Table 3 microorganisms-08-01859-t003:** The association between seroreversion and antigen-specific antibodies in the immunoblot among 215 individuals who tested positive in a serosurvey in 1997–1999 and were retested in 2008–2011.

Cumulative Number of Reactive Antigens or Specific Antigen	Number of Seropositive Participants in 1997–99	% of all Seropositive	Number of Seroreversions	% of All Seroreversions	*p*-Value ^$^	Univariable Analyses		Multivariable Analyses ^§^	
						**OR**	**95% CI**	**OR**	**95% CI**
OspC	48	22	3	8	0.02	0.25	0.06–0.74	0.39	0.08–1.38
VlsE	206	96	34	90	0.03	0.25	0.06–1.04	0.39	0.08–1.99
p39	122	57	16	42	0.04	0.49	0.24–0.99	0.97	0.42–2.26
p58	134	62	12	32	<0.01	0.21	0.09–0.43	0.51	0.20–1.26
p83	91	42	4	11	<0.01	0.12	0.04–0.32	0.27	0.06–0.84
DbpA	133	62	15	40	<0.01	0.33	0.16–0.67	0.71	0.23–2.05
Nr of antigens					<0.01				
1	26	12	11	29		Ref			
2	48	22	16	42		0.68	0.25–1.83		
3	39	18	7	18		0.30	0.09–0.90		
4	41	19	1	3		0.03	0.00–0.20		
5	35	16	2	5		0.08	0.01–0.36		
6	26	12	1	3		0.05	0.00–0.32		
Total	215		38						

^§^ The multivariable model included listed bands as well as the optical density measured in 1997–1999 ^$^ Chi-squared or Fisher’s exact test. Ref: reference category.

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
