# Peer review of "Dynamics of Borrelia burgdorferi-Specific Antibodies: Seroconversion and Seroreversion between Two Population-Based, Cross-Sectional Surveys among Adults in Germany"

_microorganisms, 2020, doi:10.3390/microorganisms8121859_

Round 1

Reviewer 1 Report

The manuscript entitled, ‘Dynamics of Borrelia burgdorferi specific antibodies: Seroconversion and seroreversion between two population-based, cross-sectional surveys among adults in Germany’ by Woudenberg et al. documents seroprevalence and seroreversion among participants from Germany in two studies. Authors show association of the level of seroreversion with the age of individuals, and with serological response to specific antigens of Borrelia burgdorferi using the standard two-step serological assays for Lyme disease. Overall, it is an interesting study and provides insight into long-term persistence of serological response in individuals. Addressing the following points/concerns will clarify some points and further strengthen this manuscript.

  1. Line 30-32. These sentences can be presented in a more direct manner. Double negative perspective dilutes the effect of this sentence. For example, an alternative sentence can be:

Serological response against p83 was associated with the lowest level of seroreversion in individuals in this study.

  1. Lines 118, 125. PBi is a Borrelia garinii strain (not B. bavariensis) and is often used by researchers as the Type strain of B. garinii in their experiments.
  2. Lines 180-187. It is advisable to define various towns as urban or rural for readers who are not familiar with the geography of Germany.
  3. In methods and under seroreversion heading, authors should define OD threshold in ELISA that they considered to define seropositive or seronegative results.
  4. Lines 219-223. Please modify this sentence. As a suggestion: Seroreversion showed lowest association with seroreactivity with the p83 antigen in individuals and highest when VlsE antigen was used in serological assays. Reactivity of antibodies of participants with multiple B. burgdorferi antigens correlated with their diminished seroreversion rate. Lines 253 to 257 are written in the most straightforward manner to describe these results.
  5. Define what does Ref indicate in Table 1.

Author Response

  1. Line 30-32. These sentences can be presented in a more direct manner. Double negative perspective dilutes the effect of this sentence. For example, an alternative sentence can be: Serological response against p83 was associated with the lowest level of seroreversion in individuals in this study.

Response: We thank the reviewer for this suggestion to increase the clarity of the manuscript. We rewrote the sentence to remove the double negative: Participants with a strong response against antigen p83 had the lowest odds to seroconvert.

  1. Lines 118, 125. PBi is a Borrelia garinii strain (not B. bavariensis) and is often used by researchers as the Type strain of B. garinii in their experiments.

Response: we appreciate the attention the reviewer has paid to our manuscript. With regard to terminology, we choose to use the abbreviation PBi as previously suggested by Margos et al. (International Journal of Systematic and Evolutionary Microbiology (2013), 63, 4284–4288). Prior to that study, B. bavariensis was termed B. garinii OspA-type 4 strain PBi (Baranton et al., 1992).

  1. Lines 180-187. It is advisable to define various towns as urban or rural for readers who are not familiar with the geography of Germany.

Response: In order to make it more conceivable for readers not familiar with the geography of Germany, we added more context by adding the number of habitants. In addition, we made it more obvious that we are discussing federal states by explicitly stating federal states twice.

  1. In methods and under seroreversion heading, authors should define OD threshold in ELISA that they considered to define seropositive or seronegative results.

Response: We used a floating cut-off as specified in the methods (line 121-122). As a result, the threshold deviated between different assays. However, as can be seen from Figure 2, lowest OD found positive are around 1.2 – 1.3

  1. Lines 219-223. Please modify this sentence. As a suggestion: Seroreversion showed lowest association with seroreactivity with the p83 antigen in individuals and highest when VlsE antigen was used in serological assays. Reactivity of antibodies of participants with multiple B. burgdorferi antigens correlated with their diminished seroreversion rate. Lines 253 to 257 are written in the most straightforward manner to describe these results.

Response: We would like to thank the reviewer for providing concrete improvements. To prevent using the term inversely, we have changed seroreversion into “testing positive in 2008-11” in line 219. We hope this removes ambiguity.

  1. Define what does Ref indicate in Table 1.

Response: we added the meaning of Ref (reference category) to the caption of table 1, 2, and 3.

Reviewer 2 Report

Woudenberg et al. presented an original study about the dynamics of Bb specific antibodies in German population, based on two surveys (1997-1999 and 2008-2011). The originality of this study is the examination of seroconversion and seroreversion rates in this population.

Minor comments : 

Please give the name of the bacteria and ticks in italics.

line 46: Reference 3 is not useful

line 52: yearly incidence for Ireland in the reference 5 is 0.6 and not 0.5

line 77: seroreversion

line 155 : 6,957 from DEGS ? 7077 in supplemental Figure 2. Please clarify

lines 159-160: 487 ? 485 in supplemental Figure1. 766 ? 756 in supplemental Figure 2. Please clarify.

3.5 Seroreversion: This point is very interesting. Are the "seroreverted" patients those with equivocal immunoblot in the first study ? If yes, this could be a potential bias , as these patients could be false positive in the first study and detected as seronegative in the second study.

line 275: seroconversion rate is 0.45%

Author Response

Please give the name of the bacteria and ticks in italics.

Response: We fully agree, we changed accordingly.

line 46: Reference 3 is not useful

 Reference 3 is a publication of a prospective population-based study and still after years provides best evidence on the incidence of different Lyme borreliosis manifestations. During a thorough awareness campaign, the majority of cases found concerned patients with an erythema migrans only (89%). Therefore, we think this publication supports our statement that erythema migrans is the most common presentation of a B. burgdorferi infection.

line 52: yearly incidence for Ireland in the reference 5 is 0.6 and not 0.5

Response: we really appreciate the critical appraisal of our manuscript, including correct use of referencing. It is indeed 0.6 and not 0.5. We changed it in the manuscript.

line 77: seroreversion

Response: We changed seroreverson into seroreversion.

line 155 : 6,957 from DEGS ? 7077 in supplemental Figure 2. Please clarify

Response: For 120 of the participants in the second survey, we were unable to estimate a weighting coefficient, which is required for a representative seroprevalence estimate of Germany. For the estimation of the seroprevalence, we used information from 6957 participants. We therefore updated supplemental Figure 2 accordingly, and corrected 766 into 738 (line 164).

lines 159-160: 487 ? 485 in supplemental Figure1. 766 ? 756 in supplemental Figure 2. Please clarify.

Response: We appreciate the effort of the reviewer to thoroughly check the figures. In this case, the numbers are correct as the number of seropositive participants is estimated by the sum of three groups (these groups can be identified by the word seropositive). The reviewer overlooked the seropositive individuals coming from an equivocal ELISA but positive line blot.

3.5 Seroreversion: This point is very interesting. Are the "seroreverted" patients those with equivocal immunoblot in the first study ? If yes, this could be a potential bias , as these patients could be false positive in the first study and detected as seronegative in the second study.

Response: in line with the reviewer, we also find the topic of serology and seroreversion very interesting. As we applied identical testing algorithm in both studies with identical tests, testing positive in both surveys had identical probabilities. We report that among participants that had tested positive in survey 1, those that turned out to be negative in survey 2 had on average lower levels of antibodies in survey 1. As higher levels of antibodies (OD in ELISA) were also associated with reactivity to a higher number of antigens, it shouldn’t come as a surprise that “seroreverters” started with less antibodies and less reactivity to antigens than “non-seroreverters”. We consider this as waning seropositivity. We think those study participants exhibit a weaker response in survey 1 but we do not see this as a bias in sense of a systematic distortion of a measurement.

line 275: seroconversion rate is 0.45%

Response: Thanks for identifying this important error. This is important. We meant to say the seroreversion rate, we changed seroconversion into seroreversion.